# Climatic, topographic, and groundwater controls on runoff response to precipitation: evidence from a large-sample data set

Zahra Eslami[1], Hansjörg Seybold[1], James W. Kirchner[1,2,3]

[1]Dept. of Environmental Systems Science, ETH Zurich, Zurich, Switzerland

[2]Swiss Federal Research Institute WSL, Birmensdorf, Switzerland

[3]Dept. of Earth and Planetary Science, University of California, Berkeley, CA, USA

*Correspondence to*: James W. Kirchner (kirchner@ethz.ch)

**Abstract.** Understanding the factors that influence catchment runoff response is essential for effective water resource management. Runoff response to precipitation can vary significantly, depending on the dynamics of hillslope water storage and release, and on the transmission of hydrological signals through the channel network. Here, we use Ensemble Rainfall-Runoff Analysis (ERRA) to characterize the runoff response of 189 Iranian catchments with diverse landscapes and climates. ERRA quantifies the increase in lagged streamflow attributable to each unit of additional precipitation, while accounting for nonlinearities in catchment behavior. Peak runoff response, as quantified by ERRA across Iran, is higher in more humid climates, in steeper and smaller catchments, and in catchments with shallower water tables. The direction and approximate magnitude of these effects persist after correlations among the drivers (e.g., deeper water tables are more common in more arid regions) are accounted for. These findings highlight the importance of catchment attributes in shaping runoff behavior, particularly in arid and semi-arid regions, where climatic variability and groundwater dynamics are crucial factors in sustainable water resource management and effective flood risk mitigation.

## 1 Introduction

Runoff generation is influenced by the interaction of different processes which vary according to climate conditions and catchment properties (Zillgens et al., 2007). Investigating catchment hydrological responses to precipitation events can provide insights into the governing factors that control streamflow generation (von Freyberg et al., 2018).

Topography plays a significant role in rainfall-runoff responses (Beven and Kirkby, 1979; Hernandez et al., 2003; Zevenbergen and Thorne, 1987; Inaoka et al., 2020), with larger catchment areas often experiencing overland flow once the land becomes saturated, leading to a substantial increase in runoff (Inaoka et al., 2020). Although gravity influences flow along topographic

and hydrological gradients, it is difficult to fully characterize dynamic runoff-generation processes using just topographic
measures such as slope, drainage area, and unit contour lengths (Mirus and Loague, 2013).
The relationship between precipitation and potential evapotranspiration, as quantified by the aridity index (FAO and UNESCO,
1977), is another critical factor influencing runoff behavior. Previous studies have shown that runoff response correlates with
the climatic aridity index (Saft et al., 2016; Barrientos et al., 2023; Matanó et al., 2024). Aridity's effects on runoff response,
as identified by various studies, can be attributed to several factors. Lower precipitation, higher temperatures, and increased
potential evapotranspiration reduce soil moisture, altering the connectivity of drainage networks with their surrounding
landscapes. Several studies (e.g. Eltahir, and Yeh, 1999; Arora, 2002; Ragab and Prudhomme, 2002; Van De Griend et al.,
2002; Rockström et al., 2010; Saft et al., 2016, Barrientos et al., 2023) show that these climatic factors reduce catchments'
water storage (such as soil moisture and groundwater), making the landscape less responsive to subsequent precipitation.
Additionally, vegetation cover and root zone dynamics may play a role in modulating runoff generation by altering infiltration
rates and the distribution of soil moisture (Gao et al., 2024).
A significant amount of runoff is generated by subsurface processes including groundwater flow and subsurface stormflow
(Bronstert et al., 2023; Jasechko, 2016). In many landscapes, precipitation must initially replenish near-surface storage before
runoff occurs (Tromp-van Meerveld and McDonnell, 2006). This process is common in arid regions with a strong surface
water-groundwater connection and deep soil layers, highlighting the relationship between changes in rainfall-runoff behavior
and catchment characteristics during dry periods (Matanó et al., 2024).
In Iran as in other arid and semi-arid environments, significant reductions in surface water availability have led to a growing
dependence on groundwater, particularly in the southern, central, and eastern regions (Nabavi, 2018; Ashraf et al., 2021; Noori
et al., 2021; Maghrebi et al., 2023). Over-extraction of groundwater can also affect runoff response in these regions (Safdari
et al, 2022). In environments where groundwater plays a significant role in maintaining streamflow during dry periods, a
declining water table can lead to reduced groundwater discharge into rivers (Jasechko et al., 2021). As a result, surface runoff
may become more erratic, and more sensitive to climate variations, emphasizing the need to consider both groundwater
dynamics and climate change in managing streamflow variability (Botter et al., 2013).
Here we investigate the runoff response of catchments across Iran, using Ensemble Rainfall-Runoff Analysis (ERRA:
Kirchner, 2024). ERRA is based on recently developed methods for estimating impulse response functions in nonlinear,
nonstationary, and heterogeneous systems (Kirchner, 2022). It is a data-driven, nonparametric, and model-independent
approach for quantifying rainfall-runoff relationships across various time lags.
Although considerable progress has been made in elucidating the factors that influence runoff response, a comprehensive
understanding of how topographic, climatic, and hydrological variables interact to shape runoff response remains elusive.
While several studies (e.g., Merz et al., 2006; Norbiato et al., 2009; Tarasova et al., 2018; Zheng et al., 2023) have explored
the controls of variable runoff response in temperate climates, such investigations are notably absent in arid environments,
underscoring the unique focus of this study in bridging this gap. Although many studies have focused on individual drivers,
the interactions between them and their combined impact on runoff response are still not fully understood. Specifically, our
analysis addresses the following questions: (1) How do topographic, climatic, and groundwater variables, and their
interactions, influence runoff response in catchments, and (2) How do variations in groundwater depth influence runoff
response when accounting for other relevant factors such as slope, or catchment size?
In this study, we apply ERRA to analyze how runoff response is influenced by several interacting factors—including
groundwater depth, aridity index, slope, and catchment area—across Iran's diverse climatic and topographic regions.

## 2 Methods

### 2.1 Study sites

Iran has a diverse climate due to its varied topography and geography. The country's climate is primarily arid to semi-arid,
with 35.5% of its land classified as hyper-arid, 29.2% as arid, and 20.1% as semi-arid. A further 5% of the country has a
Mediterranean climate, while the remaining areas, located near the Caspian Sea where rainfall is more abundant, are classified
as humid or hyper-humid (Ashraf et al., 2021). The average annual precipitation across Iran is about 240 mm; however, in the
northern provinces near the Caspian Sea, rainfall can exceed 1,800 mm annually (World Bank, n.d.). In contrast, the central
and eastern regions of Iran receive as little as 50 mm of rainfall. Potential evaporation also varies widely, from 500 mm
annually in the northwest to 3,750 mm in the southern desert regions, exceeding rainfall by a factor of 75 on annual averages.
The country's temperatures vary dramatically, ranging from an average of 0°C in the northern mountains to 28°C in the south
(Maghrebi et al., 2023).

### 2.2 Dataset

Our analysis uses daily streamflow data from 1,549 active hydrometric stations provided by the Iranian Water Resources
Management Company (IWRMC, 2018). Each station is identified by a unique site code and the location of the stream gauge
is given by latitude and longitude. For each gauge, we first extracted the corresponding upstream catchment using ArcGIS's
watershed tools and SRTM's topographic data (Jarvis et al., 2008) at 90m resolution. Pour point snapping was applied to match
gauge locations to the extracted drainage networks. For 394 gauges, the DEM catchment extraction failed, resulting in no
reasonable catchment boundaries. Consequently, these gauges were excluded from our analysis, resulting in 1,155 catchments
with corresponding gauges and streamflow. Next, daily rainfall timeseries for each catchment were extracted from CHELSA's
(Karger et al., 2017) global precipitation downscaling reanalysis. Catchments with unreasonable Q/P ratios (i.e., Q/P > 0.8)
were discarded from our analysis to exclude those with potentially erroneous or unrepresentative discharge observations. In
order to minimize the impact of dams, we also excluded catchments with large dams visible on Google Earth or where dam
effects were evident in the hydrographs. Among the analyzed catchments, 47% exhibit no overlap with others, while only 27%
overlap with other catchments by more than 20% of their drainage area.
To study groundwater–surface water interactions, we used monthly time series of groundwater depth from 13,538 wells
spanning the period 2000–2018 (IWRMC, 2018). We calculated the temporal mean groundwater depth for each time series,
then averaged these values to obtain a mean depth to groundwater for each catchment. In total we found 189 catchments with
rainfall-runoff data as well as groundwater level measurements. We split  the average groundwater depth to the water table
into three distinct categories; the shallowest 25% of these catchments were classified as "shallow groundwater" with depth
ranging from 1 to 14 m (blue points in Fig. 1a), the deepest 25% were classified as "deep groundwater" with depth ranging
between 27 and 92m (red points in Fig. 1a), and the remaining 50% were classified as "intermediate groundwater" with depth
in the range of 14-27 m (yellow points in Fig. 1a).
The aridity index (AI= $P$/PET) is widely used as a proxy to compare climatic aridity across space and time (Arora, 2002;
Nastos et al., 2013; Greve et al., 2019; Zomer et al., 2022; Barrientos et al., 2023). To ensure consistency with our precipitation
dataset, we calculated mean AI values for each catchment using CHELSA's precipitation (P) and potential evapotranspiration
(PET) time series (Karger et al., 2017). First, we computed annual means for P and PET over the period 2000–2018. Next, we
obtained the ratio AI=P/PET. Finally, we extracted the spatial mean for each catchment. Since AI is the ratio of precipitation
to potential evapotranspiration, higher AI values indicate greater humidity.
Catchment-averaged topographic slope was calculated from the 90-meter-resolution Shuttle Radar Topography Mission
Digital Elevation Model (Jarvis et al., 2008).

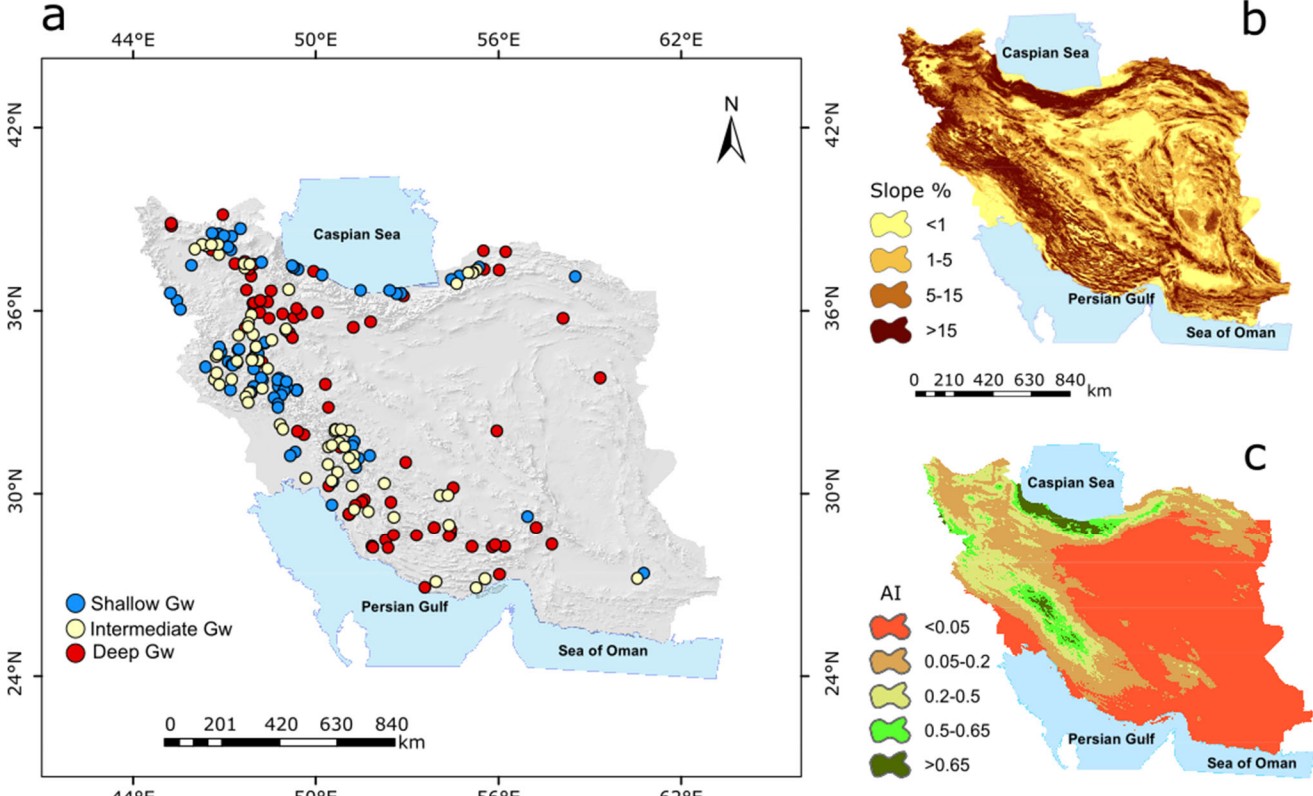

**Figure 1.** Locations of the study catchments, color-coded by groundwater depth – shallow (<14m), intermediate (14-27m) and deep (>27m) and superimposed on a hillshade map of Iran (a), with maps of hillslope gradients (b) and aridity index, AI=P/PET (c). The study sites span widely differing climatic and topographic settings, with a wide range of groundwater depths.

**2.3 Ensemble Rainfall-Runoff Analysis (ERRA)**

Here, we examine the relationship between precipitation and streamflow using Ensemble Rainfall-Runoff Analysis (ERRA: Kirchner, 2022, 2024). ERRA is a data-driven approach that generalizes classical unit hydrograph methods to account for nonlinearity and nonstationary in hydrological response. ERRA's weighted average runoff response distributions (RRDs) measure the incremental increase in streamflow, per unit of precipitation input, over a range of lag times. With ERRA, we first estimated each catchment's nonlinear response functions (NRFs), which use piecewise linear broken-stick functions to express how streamflow response varies with precipitation intensity and time lag. Each of these NRFs had four knot points ("xknots"; Kirchner, 2024), spaced as evenly as possible between the highest and lowest precipitation values, with the constraint that each broken-stick interval contained at least 20 nonzero and non-missing values. The average of these NRFs, weighted by precipitation intensity, yields the weighted average RRDs (Kirchner, 2024). In this paper, we exclusively use weighted average RRDs; however, for simplicity, we refer to them as RRDs. The RRD quantifies the catchment's average runoff response while

still accounting for underlying nonlinearities. It also avoids overestimation biases that are inherent in many applications of
conventional regression-based unit hydrograph methods to nonlinear systems (see Sect. 3.4 of Kirchner, 2024). Streamflow
can respond to precipitation over various time scales, typically reaching a peak within minutes, hours, or days, followed by a
recession that can last from days to months. Here, we studied lags up to 10 days. This was sufficient to capture every
catchment's peak runoff response (which often came during the same day that precipitation fell, or the first day afterward) and
recession back toward base flow.
We used ERRA's robust estimation option to reduce the influence of any outliers in our source data. Using the approach
outlined here, we generated runoff response distributions from our daily time series of precipitation and streamflow, and
calculated the peak heights of these RRDs for all catchments. Figure 2 shows precipitation and streamflow time series for three
example catchments with shallow groundwater levels and three catchments with deep groundwater levels. The corresponding
RRDs for lags up to 10 days are shown in Fig. 3.

### 2.4 Factors affecting runoff response distributions (RRDs)

Relationships between RRD peak height and groundwater depth, aridity index, catchment slope, and catchment area were
assessed using rank correlation and regression analyses. Spearman rank correlation coefficients ($\rho$) were used as robust
measures of monotonic relationships between RRD peak height and the four climatic, hydrologic and topographic drivers (Fig.
4). We also accounted for the confounding effects of correlations among the different drivers using partial regression leverage
plots (Fig. 5). These leverage plots measure how much the RRD peak height would change, per unit change in each of the
drivers, if the other drivers were held constant; they also facilitate the identification of individual points that disproportionately
affect the results (Cook and Weisberg, 1982; Hoaglin and Welsch, 1978; St. Laurent and Cook, 1992; Wei et al., 1998; Wright
et al., 2019).

### 3 Results and discussion

### 3.1 Runoff response distributions (RRDs)

The impulse response of rainfall to runoff can be summarized using RRDs calculated from ERRA (Fig. 3). Runoff response
distributions express how the runoff response to one unit of precipitation is distributed over time. At our 189 study catchments,
runoff response typically peaks within the first day following precipitation input and then rapidly declines, lasting about 2-3
days. The results indicate that RRD peak heights are generally higher in catchments with shallow groundwater compared to
those with deep groundwater, as illustrated in Fig. 2, which shows example comparisons between rainfall-runoff time series
for three catchments with shallow groundwater and three with deep groundwater.
Our results show that catchments with shallower groundwater, particularly in western and northern Iran, tend to have higher
average RRD peak heights. Shallow groundwater is common in Caspian Sea catchments, where many of Iran's permanent
rivers are located, and in western regions, where high precipitation keeps groundwater close to the surface, aiding runoff. In
contrast, more arid areas have deeper groundwater that is less connected to the surface, with infiltration to groundwater being
limited by evaporative demand.

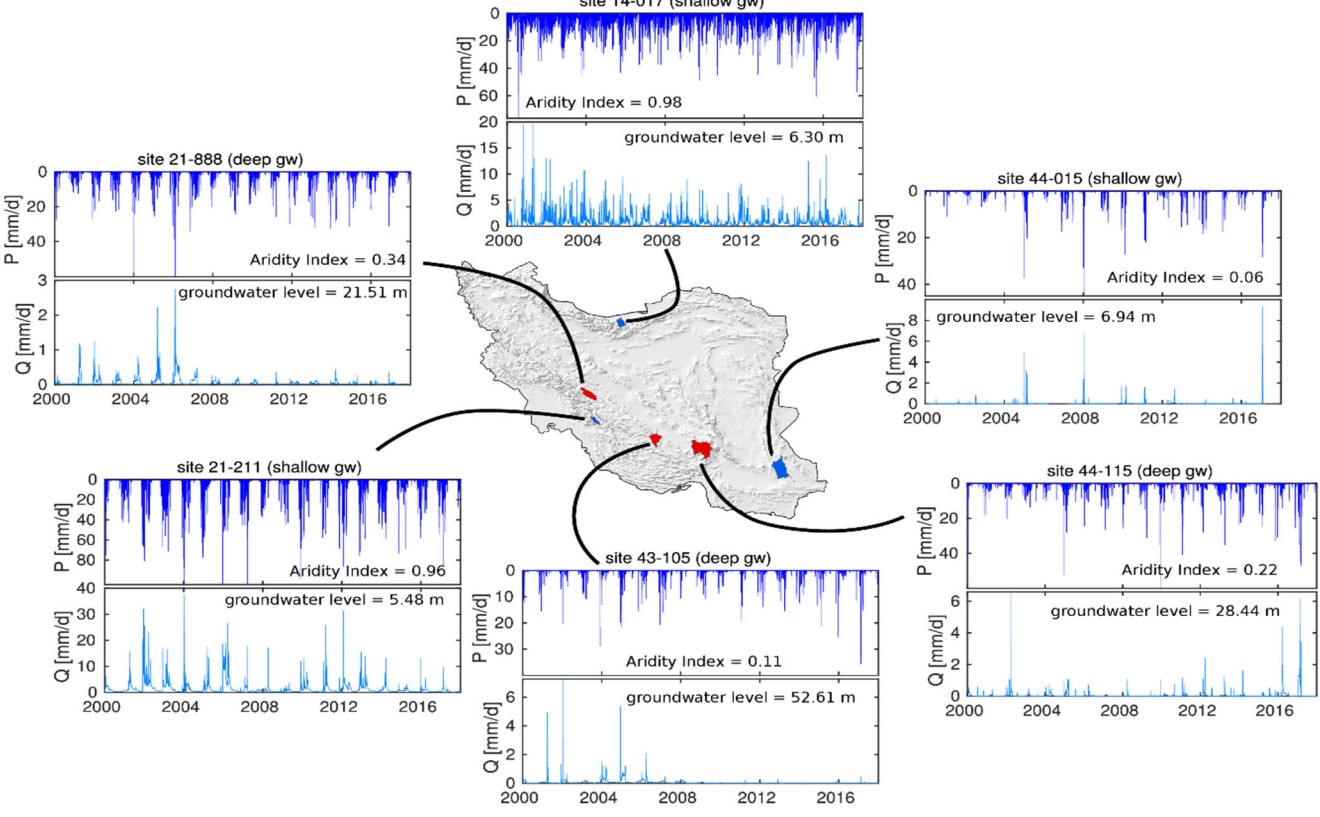


**Figure 2. Time series of precipitation and stream discharge for 3 catchments with shallow groundwater (in blue on map) and 3 catchments with deep groundwater (in red on map). Note that axis scales vary so that each catchment's behavior is visible. These example time series reflect climatic differences across Iran; at the relatively humid Caspian Sea coast (site 14-017), precipitation and runoff events are frequent, whereas at the Persian Gulf & Sea of Oman catchment (site 21-211), precipitation and streamflow are strongly seasonal, with long dry periods in summer, and at the Central Plateau catchment (site 44-015), precipitation is highly episodic, yielding infrequent and brief runoff events. Catchments with deeper groundwater tend to have lower and more episodic stream flows (compare sites 21-888 and 21-211, for example). Many catchments with deeper (and declining) groundwater (e.g., 21-888 and 43-105) exhibit visually obvious decreases in streamflow, but some others (e.g., 44-115) do not. Time series for the same sites, but over shorter time spans to reflect the details more clearly, are shown in Supplementary Figure S1.**

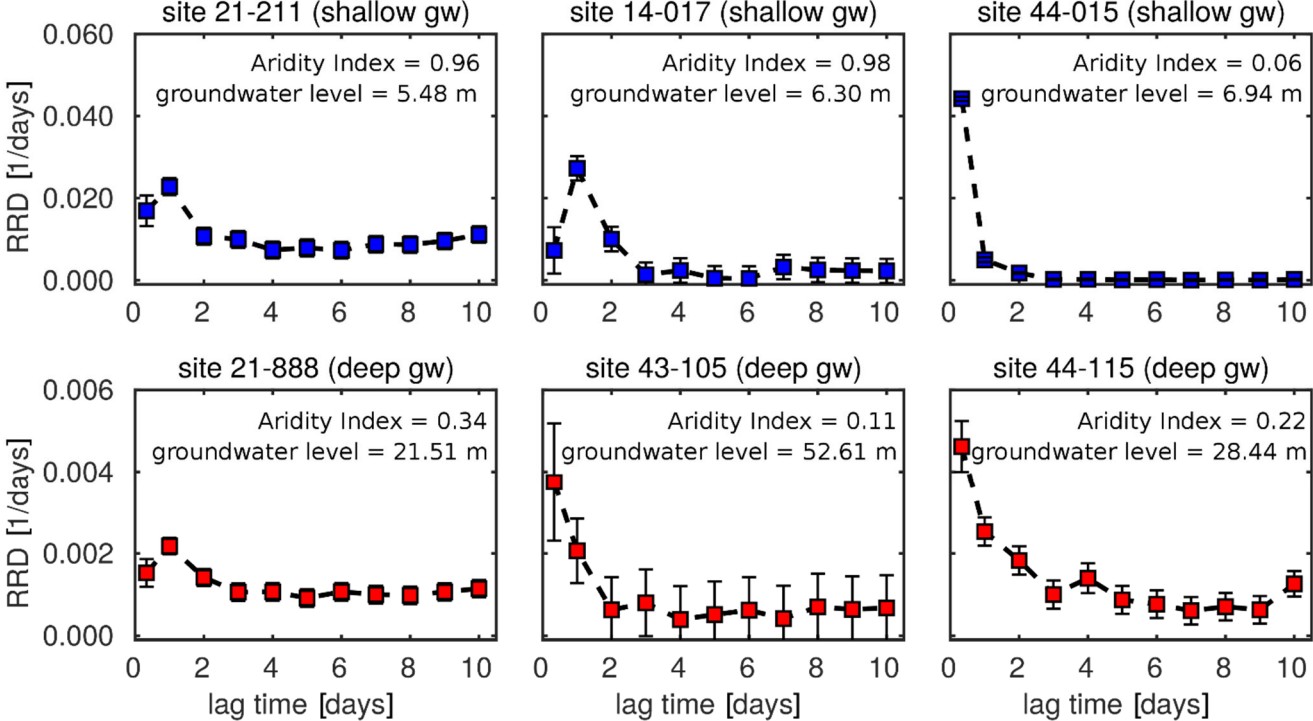

164

**Figure 3. Runoff response distributions (RRDs) for the six example catchments shown in Fig. 2. Note the factor-of-10 difference in the axis scales for the catchments with shallow groundwater (top row) versus those with deep groundwater (bottom row). Runoff response typically peaks the same day that precipitation falls, or one day after, and decays away within the next two days. Sites with shallow groundwater (top row) exhibit much stronger runoff response than those with deep groundwater (bottom row; note different axis scale).**

### 3.2 Factors influencing RRD peak height

Comparisons of RRD peak height and explanatory variables across catchments revealed a negative Spearman rank correlation coefficient ($\rho$ = -0.22, p<0.001) between RRD peak height and groundwater depth (Fig. 4). This indicates that deeper groundwater levels are associated with smaller RRD peak heights, while catchments with shallower groundwater exhibit higher runoff peaks in response to precipitation. In regions with shallow groundwater levels, the subsurface's capacity to store excess water is limited, leading to higher near-surface runoff during intense rainfall events. Rapid saturation of near-surface layers can also contribute to increased overland flow (e.g., Steenhuis et al., 2005). Conversely, deeper groundwater levels enhance subsurface water retention, promoting infiltration and making streamflow less responsive to precipitation.

Differences in runoff behavior between areas with shallow and deep groundwater can also be attributed to subsurface flow paths. In regions with shallow water tables, translatory flow dominates, quickly displacing water stored in the soil (Hewlett

and Hibbert, 1967), resulting in sharp runoff response peaks. In contrast, deeper groundwater allows for deeper infiltration of rainwater, bypassing intermediate layers and delaying saturation (Floriancic et al., 2024), thus leading to lower runoff response peaks and more gradual hydrological responses. However, the relatively weak correlation suggests that while groundwater depth may influence RRD peak height, it is unlikely to be the primary controlling factor in the study area.

The correlation analysis revealed a positive Spearman correlation ($\rho$=0.43, p<0.001) between RRD peak height and topographic slope, suggesting that RRD peak height is generally higher in steeper terrain. This finding is consistent with hydrological theory, as stronger topographic gradients accelerate runoff, contributing to sharper and more pronounced peaks (Inaoka et al., 2020). The observed correlation underscores the importance of topographic features in catchment hydrology and suggests that slope should be considered when assessing runoff potential in similar landscapes.

Additionally, a negative Spearman correlation ($\rho$=−0.21, p<0.001) was found between RRD peak height and catchment area, indicating that larger catchments tend to have lower RRD peak heights. These smaller RRD peak heights may result from dispersion of runoff peaks during transmission through the drainage network, or from the superposition of runoff peaks generated at different distances from the outlet (and thus lagged by different amounts before they reach the gauging station). Larger catchments also may encompass more varied topographic and soil characteristics, leading to a wider variety of sub-catchment runoff responses which are combined at the outlet. Nonetheless, the weak correlation suggests that catchment area alone does not dictate RRD peak height, implying a more complex interaction among factors that influence runoff behavior.

Our analysis revealed a positive correlation between climatic aridity index and RRD peak height ($\rho$=0.27, p<0.001). This observation aligns with the research of Barrientos et al. (2023), who reported that runoff response is sensitive to variations in aridity. However, the modest correlation between AI and RRD peak height suggests that AI is only one of several factors, including climatic, ecological, geographic, geological, and anthropogenic drivers, that influence runoff behavior (Van Dijk et al., 2013; Schewe et al., 2014; Barrientos et al., 2023). In arid regions (low AI), for example, groundwater wells are generally also deeper, while in more humid regions (higher AI), wells tend to be shallower (Fig. S2). The catchments near the Caspian Sea (AI > 0.65), for example, frequently display shallow groundwater tables and higher RRD peak heights. Similarly, catchments in other humid regions of Iran (AI between 0.5 and 0.65) also show shallower groundwater and higher RRD peaks. By contrast, in more arid regions, deeper groundwater levels, limited surface connectivity, and higher evaporative demand contribute to reduced recharge and lower RRD peaks.

Readers should note that because the RRD quantifies the increase in streamflow per unit of precipitation, it normalizes for the differences in precipitation amounts between humid and arid catchments. Thus, the higher RRD peaks in humid catchments do not reflect the fact that precipitation amounts tend to be higher there. Instead, the higher RRD peaks imply that humid catchments generate more streamflow, per unit of precipitation, in response to rainfall, with the result that peak runoff response increases more-than-proportionally to precipitation inputs.

To further examine these patterns, we computed correlations between mean specific discharge (discharge per unit basin area)
and the catchment attributes described above. This analysis revealed broadly similar relationships to those observed for the
RRD (Figure S3).
We used partial regression leverage plots to better understand the significance and relative influence of each driver for RRD
peak height (Fig. 5). Figure 5 compares the leverage of log-transformed RRD peak height against the leverage of our four log-
transformed explanatory variables (groundwater depth, aridity index, catchment area, and slope). Leverage plots show the
effects of each driver, with the linear effects of the other drivers removed. The results shown in Fig. 5 are broadly similar to
those shown in Fig. 4, except for AI, which shows a reversed, but statistically insignificant trend with $p>0.05$. All other
variables are statistically significant drivers of RRD peak height with $p<0.001$. Topographic slope emerges as the strongest
control on RRD peak height, followed by catchment area and groundwater depth (see effect tests in Table 1).
These findings highlight the close relation between aridity and groundwater depth. In humid regions (i.e., with high AI), where
water tables are closer to the surface, limited subsurface storage capacity reduces the ability to absorb rainfall, potentially
increasing near-surface runoff and accelerating hydrological response (Erdbrügger et al., 2023). Conversely, in arid regions
(i.e., with low AI), groundwater tends to be deeper, and subsurface layers often retain more capacity to store rainfall. This
leads to lower and more delayed runoff peaks, as more water is absorbed into the unsaturated layers or lost through evaporation
(Condon et al., 2020).
While this study primarily focuses on climatic, groundwater, and topographic controls, other factors, such as geological
heterogeneity, may also contribute to variability in runoff response. Differences in bedrock permeability and soil properties
can influence infiltration and water storage, underscoring the potential role of subsurface properties in shaping hydrological
processes (Izadi et al., 2020). Although detailed geological analysis was beyond the scope of this study, future research
could explore how geological variability interacts with climatic and topographic factors to refine our understanding of runoff
generation across diverse landscapes.

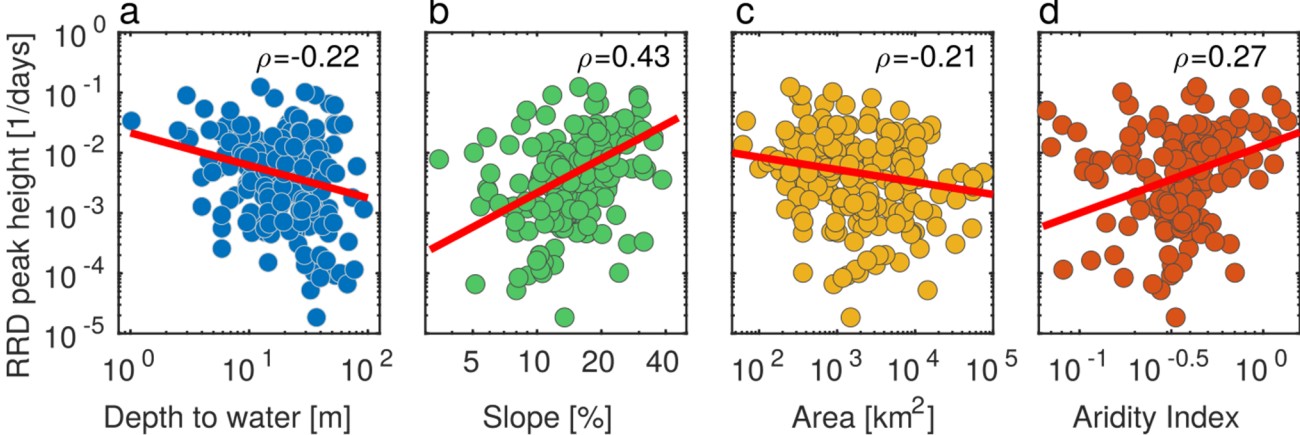

Figure 4. Scatterplots relating RRD peak height to four catchment attributes (all axes are logarithms): depth to groundwater (a), mean topographic slope (b), drainage area (c), and aridity index (d). All Spearman rank correlations ($\rho$) are statistically significant at p<0.001.

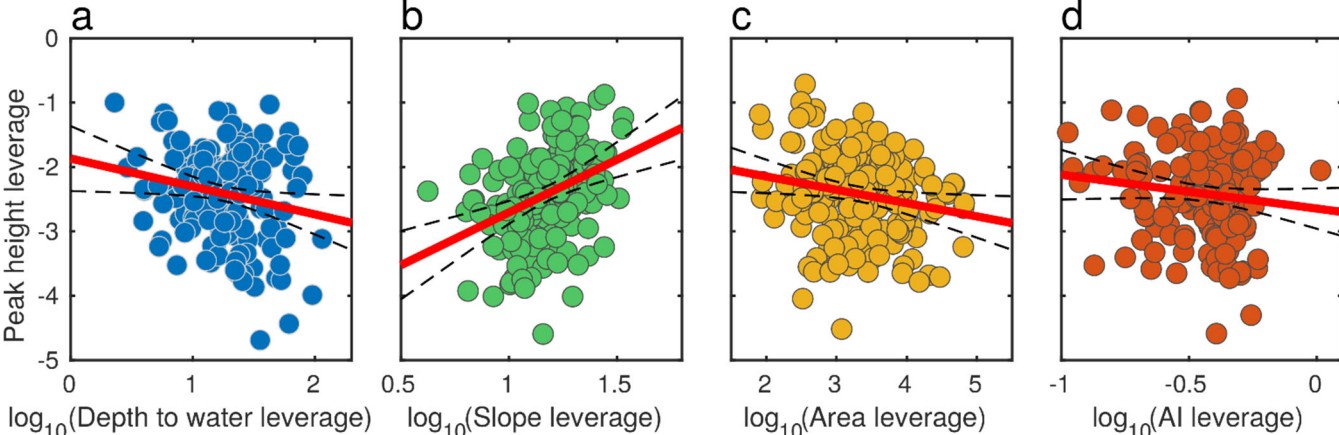

Figure 5. Leverage plots relating RRD peak height to four catchment attributes: depth to groundwater (a), mean topographic slope (b), drainage area (c), and aridity index (d). All axes are logarithms. Leverage plots show the effects of each variable, with the linear effects of the other three variables removed. Red lines show multiple linear regression slopes, and dashed lines show p=0.05 confidence bounds. All attributes are statistically significant at p<0.01 except $\log_{10}(AI)$, for which p=0.05 (Table 1).

**Table 1. Partial regression results: effects of groundwater level, topography and climate on $\log_{10}$(RRD peak height)**

| variable | estimate | standard error | t ratio | prob>\|t\| | effect test sum of squares |
|---|---|---|---|---|---|
| $\log_{10}$(groundwater depth) | -0.43 | 0.16 | -2.72 | 0.0072 | 3.07 |
| $\log_{10}$(mean slope) | 1.64 | 0.31 | 5.25 | <0.0001 | 11.45 |
| $\log_{10}$(basin area) | -0.20 | 0.07 | -2.79 | 0.0058 | 3.24 |
| $\log_{10}$(AI) | -0.53 | 0.27 | -1.95 | 0.0527 | 1.58 |

## Conclusions

This study examines the complex relationships between topographic, climatic, and hydrological factors in shaping peak runoff response to precipitation inputs, as quantified by runoff response distribution (RRD) peak heights estimated by ensemble rainfall-runoff analysis (ERRA) for catchments across Iran (Figs. 1-3). The findings reveal that topography and climate are important controls on RRD peak height (Fig. 4), with topographic slope being the most influential factor, followed by basin area and groundwater depth (Fig. 5, Table 1). Steeper slopes accelerate runoff, producing sharper RRD peaks, while regions with higher AI (more humid climates) tend to have stronger runoff responses per unit of precipitation due to shallower groundwater tables and limited infiltration (Fig. 4). Conversely, regions with lower AI (more arid climates) and deeper groundwater levels exhibit more subdued runoff responses due to greater subsurface water retention. Shallow groundwater enhances runoff by allowing more rapid mobilization of subsurface storage, while deeper groundwater promotes infiltration. Larger catchment areas tend to disperse runoff peaks because flows generated in different parts of the catchment reach the outlet at different times. This study highlights the importance of considering multiple interacting factors when assessing runoff behavior, particularly in arid and semi-arid regions where climate and groundwater conditions play a crucial role in shaping hydrological responses. These insights may be helpful in developing effective water resource management strategies and mitigating flood risks in vulnerable regions.

*Data availability*. The stream flow and groundwater dataset used in our analysis is available for download at https://stu.wrm.ir. Daily rainfall and potential evapotranspiration time series for each catchment were obtained from the CHELSA climatology dataset (Karger et al., 2017).

*Author contributions*. ZE, HJS and JK conceived the work. The analysis was carried out by ZE and HJS under supervision of JWK. All authors collaboratively discussed the methodology, interpreted the results. ZE wrote the manuscript with inputs from all co-authors.

*Competing interests*. The authors have declared that there are no competing interests.

*Disclaimer*. Publisher's note: Copernicus Publications remains neutral with regard to jurisdictional claims made in the text,
published maps, institutional affiliations, or any other geographical representation in this paper. While Copernicus Publications
makes every effort to include appropriate place names, the final responsibility lies with the authors.

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
