# Peer review of "Climatic, topographic, and groundwater controls on runoff response"

_EGUsphere, 2025_

## Author Response (AR1)

**We thank the reviewers for these helpful and insightful comments. Below we provide a point-by-point response to the individual questions/suggestions.**

**The reviewers' comments are marked in normal font and our responses are in bold.**

**Responses to Reviewer 1:**

It was a please to read the manuscript on "Climatic, topographic, and groundwater controls on runoff response to precipitation: evidence from a large-sample data set". The study analyzes the potential controls of the effectiveness of runoff generation per unit of precipitation over a large number of catchments in Iran using recently developed Ensemble Rainfall-Runoff Analysis. The authors examine catchment area, aridity, and slope and groundwater depth as potential controls. The study provides important insights on the controls of runoff generation in arid and semi-arid environments.

**Thank you!**

The manuscript is well-written and structured. I suggest to additionally highlight the importance of such studies in the arid environments. Moreover, I recommend a more rigorous selection of representative groundwater wells that might be the reason behind its lower importance. Please find my detailed comments below.

Detailed comments

Choice of the representative groundwater well: From the description in the manuscript, it was not clear how groundwater wells were linked with corresponding surface catchments. In my opinion the choice of a representative well is not straightforward, especially in case of complex hydrogeological settings and large surface catchments. This might also be the reason for the relatively weak correlations between runoff peak height and groundwater depth (Line 242). I suggest to use hydrogeological maps (e.g., hydraulic conductivity) to identify representative wells out of more than 13,000 wells available for the study.

**We understand that selecting a representative well is challenging, particularly in complex hydrogeological settings and large catchments. For this reason, we intentionally did not select representative wells but instead calculated the temporal mean groundwater depth for all well level time series that were available in each catchment, and averaged these to obtain a catchment mean value.**

Introduction: The novelty can be additionally highlighted in the Introduction by outlining possible differences in the controls of runoff response between arid and temperate climates. While in the temperate climates several large sample studies (e.g., Norbiato et al., 2008 (https://doi.org/10.1016/j.jhydrol.2009.06.044); Tarasova et al., 2018 (https://doi.org/10.1029/2018WR022588); Zheng et al., 2023 (https://doi.org/10.1029/2022WR033226)) investigated potential controls of variable runoff response, in the arid environments such studies are indeed missing.

**A new paragraph has been added to the revised manuscript, emphasizing the limited research on streamflow response, especially in arid landscapes. This addition can be found in lines 55-63.**

Line 33-35: A somewhat more differentiated argument could be useful here, summarizing the main findings of these large list of studies that are named here.

**In the revised manuscript (lines 34–36) we clarified that storage levels refer to both groundwater and soil moisture. We have also added appropriate references to support this terminology.**

Line 37-38: It might be worth mentioning here Tromp-van Meerveld and McDonald, 2007 (doi:10.1029/2004WR003778) here.

**Thank you very much for drawing our attention to this reference.  We have included it in the revised manuscript at line 41.**

Line 46: It might be worth mentioning here the work of Botter et al. 2013 (https://doi.org/10.1073/pnas.1311920110)

**This reference has been added to the revised manuscript in line 50.**

Line 66: Please correct the reference here.

**This is standard citation style for an un-dated reference where the author is an organization rather than an individual. We updated the reference to include the URL as well as the date accessed.**

Line 76: It is not quite clear what is meant by a "reasonable catchment"? Please clarify. Moreover, please indicate if the catchment area was provided by the corresponding authorities and if it was used to test the accuracy of the delineated catchments.

**The term "unreasonable catchment" referred to gages for which the watershed delineation process failed. We have clarified this in the revised manuscript in lines 82–83.**

In Line 78: Please motivate the choice of this dataset. Was it tested in Iran or in the comparable environments?  Please also specify its spatial resolution.

**CHELSA is a widely used global precipitation dataset at daily resolution. Although the dataset has not been specifically tested in Iran, it has been validated in a range of similar semi-arid and mountainous regions. The dataset has a spatial resolution of 30 arc-seconds (approximately 1 km), which allows for detailed analysis of rainfall patterns at a regional scale, making it ideal for extracting daily rainfall time series for each catchment in the study.**

Line 79-81: Please clarify if the Q/P criteria was used to avoid anthropogenically affected areas, or to eliminate catchments with erroneous Q and P observations.

**We used the ratio of Q/P in a first step to eliminate basins with obviously wrong hydrographs. This has been explained in the revised manuscript in lines 85–86.**

Line 85: Please clarify if only one groundwater well per catchment was used.

**Our analysis does not rely on a single groundwater well per catchment. Instead, we first calculated the temporal mean groundwater depth for every well within a catchment and then average these individual well means to derive an overall mean depth to groundwater for each catchment.**

Figure 2: I suggest to display maximum 2 years of time series. Otherwise, the differences between stations are not really visible.

**We recognize the suggestion to display a shorter time series for better visibility of station differences. However, we chose to present the entire time series to capture long-term trends, including regions where streamflow is gradually declining. Limiting the display to only two years could obscure important patterns and hydrological changes crucial to our analysis. To address the reviewer's concern, we have added a supplementary plot (Figure S1) showing a more detailed view of the time series for the period 2000–2003.**

**Responses to reviewer 2:**

I am pleased to see that this paper is a practical application of the Ensemble Rainfall-Runoff Analysis (ERRA) method, as presented in the impressive paper by Professor Kirchner, 2024 HESS. The paper is concise and well-structured, making it an enjoyable read with no redundant text or sections. It makes a valuable contribution to the understanding of runoff response in arid and semi-arid regions, particularly in Iran, by employing the innovative Ensemble Rainfall-Runoff Analysis (ERRA) method. The use of a large-sample dataset (211 catchments) spanning diverse climatic and topographic conditions enhances the generalizability of the findings.

**Thank you.**

My comments:

1. I wonder why you did not analyze whether Q/A (runoff per unit area) shows a statistically meaningful relationship with slope, groundwater depth, and aridity index. Wouldn't this provide a useful benchmark

alongside the ERRA analysis? Even if the correlations are weak, wouldn't presenting this analysis address potential reader curiosity and offer additional insight?

**Thank you very much for the suggestion. We analyzed correlations between average specific discharge (q) and the same catchment attributes that we examined with the RRD. The correlation patterns for q were broadly similar to those for RRD. These results are presented in Supplementary Figure S3. We have incorporated this information into the revised manuscript text at lines 211–213.**

2. I noticed that some of the results cannot be well explained by traditional runoff mechanisms, such as Hortonian or saturation excess runoff, as described in textbooks and literature. I wonder if you avoided discussing these mechanisms for this reason or simply felt no need to include such a discussion. Similarly, you avoided mentioning or justifying the results, particularly those related to the aridity index using the Budyko approach. I'm not sure how to best address this, but I thought it might be worth mentioning for further consideration if it aligns with the focus of the paper.

**Storm runoff mechanisms are difficult to see in daily streamflow dynamics, and an exploration of runoff mechanisms would require much more detailed information on soil moisture, groundwater dynamics, etc. In any case our main objective here is to see how runoff behavior correlates with possible drivers, not to attribute runoff behavior to specific mechanisms.**

3. While readers familiar with Professor Kirchner's excellent work (HESS 2024) may understand ERRA, those encountering it for the first time will likely find the current explanation insufficient (section 2.3).

**We have expanded our explanation of the Ensemble Rainfall–Runoff Analysis method in Section 2.3 of the revised manuscript.**

4. You classify groundwater depth into shallow, intermediate, and deep based on percentiles (25%, 50%, and 25%, respectively), but you do not provide the actual depth ranges for these categories.

**Thank you very much for this useful comment. We add the ranges corresponding to the four quartiles in the revised manuscript at lines 94–97.**

5. Could you clarify whether all 211 catchments are non-nested, or if some share nested relationships? If applicable, please explain how this was considered in the analysis.

**Thank you for raising this point. 47% of our catchments contain no overlap with other catchments, and only 27% of the analyzed catchments overlap with other catchments by more than 20% of their drainage areas. We have added this information to the text of our revised manuscript at lines 88–89.**

6. Don't you think that erosive features and geomorphologic parameters (such as drainage density) could play a significant role in shaping runoff response, and their inclusion or acknowledgment in the study would have added depth to the analysis? Currently nothing has been mentioned about them in the paper.

**We recognize that factors such as drainage density could play an important role in shaping the runoff response. However, due to the lack of detailed stream maps for the study basins, we were unable to independently estimate drainage density (DD). However, our analysis includes slope, which is a typical geomorphic variable.**

7. You could have acknowledged the potential influence of geology in your discussion. (In general, geology has been largely ignored in the paper. The word "geology" and any of its derivatives are mentioned only once, in line 194). For example, you could have noted that geological heterogeneity (e.g., variations in bedrock permeability and soil type) may contribute to variability in runoff response but was not included due to data limitations or scope constraints. Including examples from the literature, such as Izadi et al. (2020) Investigating the Effects of Lithological Units on Runoff Coefficient (A case study of 18 watersheds in three climatic regions of Iran) (in Persian, but with figures and an English abstract that clearly illustrate the impact of lithological units on runoff coefficients in Iranian watersheds), would have highlighted the importance of geology in hydrological processes.

**We agree with the reviewer that geology, infiltration coefficients and permeability have a significant effect on runoff generating processes. We have expanded our discussion on the potential effects of these geological factors, including a reference to the findings of Izadi et al. (2020), in lines 227–232 of the revised manuscript.**

8. I believe the literature review on studies involving Iran data could be more extensive, although I appreciate the paper's current concise structure.

**We now reference additional studies from Iran in the Introduction of the revised manuscript at lines 45–47.**

9. Corrections for Figure 1:

Scale Unit: The unit for the scale should be "km" (lowercase "k" and "m") instead of "Km". This follows the standard scientific notation for kilometers.

Legend Label: The word "Legend" can be deleted from the map. The legend itself (color-coded circles) is sufficient to indicate what the colors represent, and the label is redundant.

Legend Symbols: The legend in Figure 1a currently uses colorful squares to represent the groundwater depth classifications (shallow, intermediate, deep). However, the map itself uses colorful circles. The legend should be updated to use circles instead of squares to match the map symbols. This will avoid confusion.

Geographic Coordinates: The geographic coordinates around the map (latitude and longitude) are missing the degree symbol (°). The coordinates should be labeled with the degree symbol (e.g., 30°N, 50°E) to conform to standard geographic notation.

**Thank you for your detailed observations. The requested corrections for Figure 1 have been implemented,**

10. Please ensure consistency in the formatting of axis labels by using parentheses for units in Figure 2, as done in the other figures.

**Thank you very much for catching these glitches in the figures. We have revised the figures according to your suggestions, including labeling the units in square brackets to ensure consistency**.

11. In several parts of the discussion (e.g., Sections 3.1, 3.2, and Conclusion), you have used "peak" without specifying "RRD peak", which could lead to misunderstandings. For example, readers might mistakenly interpret the findings in terms of peak discharge, which could contradict general hydrological knowledge (e.g., larger catchments typically have higher peak discharges, but the study finds that larger catchments have lower RRD peaks). To avoid confusion, you should consistently use "RRD peak" instead of "peak" throughout the manuscript.

**Thank you very much for this comment. We have adjusted the terminology in our revised manuscript always using RRD peak height, when referring to the peak of the RRD curve.**

**Responses to editor's specific comments:**

Major Comment:

The analysis comprehensively addresses climatic, topographic, and groundwater controls but omits discussion of vegetation cover and root zone dynamics. As highlighted in our recent work (Gao et al., 2024), root zones play a critical role in land-surface processes, including runoff generation and evaporation. In vegetated areas, precipitation is initially intercepted by the canopy, followed by root zone infiltration to replenish soil moisture deficits before runoff occurs. In sparsely vegetated or bare regions, intense precipitation may generate runoff when rates exceed infiltration capacity—even with limited total rainfall. I suggest the authors to consider how vegetation and root zone properties might influence their conclusions, particularly given the variability in land cover across Iran's arid and semi-arid regions.

Reference:

Gao, H., et al. (2024). Root zone in the Earth system. Hydrology and Earth System Sciences, 28, 4477–4499. https://doi.org/10.5194/hess-28-4477-2024

**We appreciate the importance of this question, but unfortunately to the best of our knowledge, no root data is available for Iran, so this could not be explicitly considered in our analysis. We however added the suggested reference regarding the importance of roots for infiltration (line 38).**

Minor Comments:

Figure 1b, c: Please improve the resolution and increase the font size for better readability.

**We have redrafted the figure with the goal of consistency between the various panels.**

Lines 189–190: The text states that catchments near the Caspian Sea have "AI > 0.65" and are later compared to "wetter parts of Iran, with AI between 0.5 and 0.65." This implies AI > 0.65 represents wetter conditions, but the phrasing could be clarified to avoid ambiguity.

**Thanks, we have changed the phrasing (now at line 203) to make it clearer.**